# Empirical estimates of the mutation rate for an alphabaculovirus

**Dieke Boezen**[1]�, **Ghulam Ali**[2]☨¤, **Manli Wang**[3], **Xi Wang**[3], **Wopke van der Werf**[4], **Just M. Vlak**[2], **Mark P. Zwart**[1]*

**1** Department of Microbial Ecology, The Netherlands Institute of Ecology (NIOO-KNAW), Wageningen, The Netherlands, **2** Laboratory of Virology, Wageningen University and Research, Wageningen, The Netherlands, **3** Wuhan Institute of Virology, Chinese Academy of Sciences, Wuhan, PR China, **4** Centre for Crop Systems Analysis, Wageningen University and Research, Wageningen, The Netherlands

he These authors contributed equally to this work.
¤ Current address: Air University Multan Campus, Multan, Pakistan
* M.Zwart@nioo.knaw.nl

**Data Availability Statement:** Data and scripts are included in the supplementary material. Raw sequence data have been uploaded to Sequence Read Archive under accession PRJNA798700 (https://www.ncbi.nlm.nih.gov/sra/PRJNA798700).

## Abstract

Mutation rates are of key importance for understanding evolutionary processes and predicting their outcomes. Empirical mutation rate estimates are available for a number of RNA viruses, but few are available for DNA viruses, which tend to have larger genomes. Whilst some viruses have very high mutation rates, lower mutation rates are expected for viruses with large genomes to ensure genome integrity. Alphabaculoviruses are insect viruses with large genomes and often have high levels of polymorphism, suggesting high mutation rates despite evidence of proofreading activity by the replication machinery. Here, we report an empirical estimate of the mutation rate per base per strand copying (s/n/r) of Autographa californica multiple nucleopolyhedrovirus (AcMNPV). To avoid biases due to selection, we analyzed mutations that occurred in a stable, non-functional genomic insert after five serial passages in *Spodoptera exigua* larvae. Our results highlight that viral demography and the stringency of mutation calling affect mutation rate estimates, and that using a population genetic simulation model to make inferences can mitigate the impact of these processes on estimates of mutation rate. We estimated a mutation rate of $\mu = 1 \times 10^{-7}$ s/n/r when applying the most stringent criteria for mutation calling, and estimates of up to $\mu = 5 \times 10^{-7}$ s/n/r when relaxing these criteria. The rates at which different classes of mutations accumulate provide good evidence for neutrality of mutations occurring within the inserted region. We therefore present a robust approach for mutation rate estimation for viruses with stable genomes, and strong evidence of a much lower alphabaculovirus mutation rate than supposed based on the high levels of polymorphism observed.

## Author summary

Virus populations can evolve rapidly, driven by the large number of mutations that occur during virus replication. It is challenging to measure mutation rates because selection will affect which mutations are observed: beneficial mutations are overrepresented in virus

**Funding:** G.A. was supported by the Netherlands Fellowship Program PhD Grant No. CF7554/2011 (Nuffic, www.nuffic.nl). The funders had no role in study design, data collection and analysis, decision to publish, or preparation of the manuscript.

**Competing interests:** The authors have declared that no competing interests exist.

populations, while deleterious mutations are selected against and therefore underrepresented. Few mutation rates have been estimated for viruses with large DNA genomes, and there are no estimates for any insect virus. Here, we estimate the mutation rate for an alphabaculovirus, a virus that infects caterpillars and has a large, 134 kilobase pair DNA genome. To ensure that selection did not bias our estimate of mutation rate, we studied which mutations occurred in a large artificial region inserted into the virus genome, where mutations did not affect viral fitness. We deep sequenced evolved virus populations, and compared the distribution of observed mutants to predictions from a simulation model to estimate mutation rate. We found evidence for a relatively low mutation rate, of one mutation in every 10 million bases replicated. This estimate is in line with expectations for a DNA virus with self-correcting replication machinery and a large genome.

## Introduction

Mutation rates are of key importance for understanding and predicting evolutionary patterns, as the mutation rate modulates the mutation supply of a population [1]. Large mutation supplies can fuel rapid and repeatable adaptation [2, 3], but also increase the mutational load on a population [4]. By contrast, low mutation supplies can limit the rate of adaptation [5], but also result in a lower mutational load [4]. The impact of mutational supply depends on the topography of the fitness landscape. Small mutational supplies can have advantages for evolution on rugged fitness landscapes: although adaptation will be slower and in most cases less fit genotypes will be selected, some populations can avoid becoming trapped on local fitness peaks [6]. Mutation rates are not only relevant to understanding basic evolutionary processes, but they also impinge on real world outcomes, such as the efficacy of prophylactic or therapeutic interventions to infectious diseases [7, 8].

Viruses have high mutation rates compared to cellular life forms [7, 9, 10], with estimates of mutations per site per strand copying (s/n/r) ranging from $2 \times 10^{-8}$ for Enterobacteria phage T2 [11] to $2 \times 10^{-4}$ for Influenza A virus [12]. Whilst these high mutation rates are thought to contribute to the rapid adaptation of viruses, beneficial mutations are typically rare, as the majority of mutations are neutral or deleterious [13, 14]. Many viruses with large genomes belong to Group I of the Baltimore classification (dsDNA viruses) [15], and typically have polymerases with proofreading activity, which should enhance the fidelity of replication [16]. A general expectation is therefore that viruses with relatively large genomes have lower mutation rates [9]. An inverse relationship between genome size and mutation rate indeed has been found [9, 17]. Small genomes can tolerate higher mutation rates as a larger proportion of mutation-free genomes are generated in each round of replication, due to their small size.

The alphabaculoviruses are a large group of insect baculoviruses that have been studied because of their biocontrol and biotechnological potential [18, 19]. Alphabaculoviruses have relatively large dsDNA genomes compared to other viruses [20], and high levels of within-host genetic diversity have been documented from wild [21–23] and captive [24] insect populations. It has been suggested that baculoviruses might therefore have high mutation rates despite their large genome sizes [24]. To our knowledge, no empirical estimates of the mutation rate have been reported for any baculovirus or insect virus to date, and there are only a few estimates for other large dsDNA viruses [9].

A major challenge for making empirical estimates of mutation rates is the need to account for biases due to selection [9]. Selection will decrease the frequency of deleterious mutations,

whist it will increase the frequency of rare beneficial mutations. These opposing effects of purifying and directional selection make it problematic to derive information on mutation rates directly from mutation accumulation patterns. Many different approaches have been developed to remove the bias introduced by selection [7, 9]. For example, some studies considered the frequency of lethal mutations in a population, since these variants cannot replicate autonomously and therefore represent a snapshot of genetic variation [25]. Others have evolved viruses in hosts expressing a viral gene, and then restricting their analysis to the sequence of this redundant viral gene [26]. Another strategy reported recently has been to incorporate fluorescent markers with inactivating mutations into a viral genome, and then performing fluctuation tests based on recovering fluorescence [12]. Finally, others have setup experiments with demographic conditions that remove variation while limiting the role of selection, in combination with high-fidelity high-throughput sequencing [27].

In the current study, we report the first empirical estimate of mutation rate for a large dsDNA insect virus, the alphabaculovirus Autographa californica multiple nucleopolyhedrovirus (AcMNPV). To ensure selection did not bias our estimates, we analyzed virus populations carrying a large, nonfunctional genomic insert that was stably maintained [28], exploiting the genomic stability of the Group I viruses [29]. For our analysis, we assumed that mutations in this region are neutral due to the absence of known viral genes and regulatory sequences, and verified this assumption. We also developed a population genetics simulation model to estimate mutation rates from empirical data that incorporates the effects of population bottlenecks and different modes of virus replication on the occurrence and maintenance of mutations. Using this approach, we made robust estimates of the mutation rate, for the first time for a baculovirus.

## Results and discussion

### Serial passage and detection of mutations

To estimate the mutation rate of a large dsDNA virus, we experimentally evolved a variant of alphabaculovirus AcMNPV containing a stable, non-functional genomic region. The AcMNPV variant used was a so-called bacmid: an infectious clone that also contains the AcMNPV genome (~134 kb) for the E2 variant [28]. It also contains bacterial sequences that enable propagation as a low copy number plasmid in *Escherichia coli* (~12.5 kb) and the acceptance of expression cassettes by transposition [28]. The specific variant used here contains an expression cassette from the pFastBac-Dual vector to restore expression of complete and functional polyhedrin (Fig 1). We consider the inserted bacterial sequences to be non-functional and therefore neutral in insects, except for the polyhedrin promoter and open reading frame (ORF) sequences derived from the pFastBac Dual vector. This renders two neutral sequences with a combined length of 11,646 bp flanking the polyhedrin gene, and the former can be studied in a mutation accumulation experiment (Fig 1). By contrast, the remainder of the AcMNPV genome is intact and unaltered, making this bacmid-derived virus a good representative of an alphabaculovirus. We will hereafter refer to this bacmid-derived AcMNPV variant with restored polyhedrin expression as "BAC", and the two neutral sequences it contains as the "neutral region". The virus could be reconstituted from the infectious clone by transfection of the BAC genome into fourth instar (L4) *Spodoptera exigua* (Hübner) (see Materials and Methods). Mutations were allowed to accumulate across the BAC genome by experimentally evolving five replicate BAC lineages (referred to as lineages A, B, C, D and E) for five passages in *S. exigua* L2. For each replicate lineage, passaging was performed in five larvae exposed to a high viral dose of occlusion bodies (OBs) sufficient to kill all larvae. Upon death larval cadavers were collected and pooled prior to the isolation of OBs, which were used to inoculate larvae

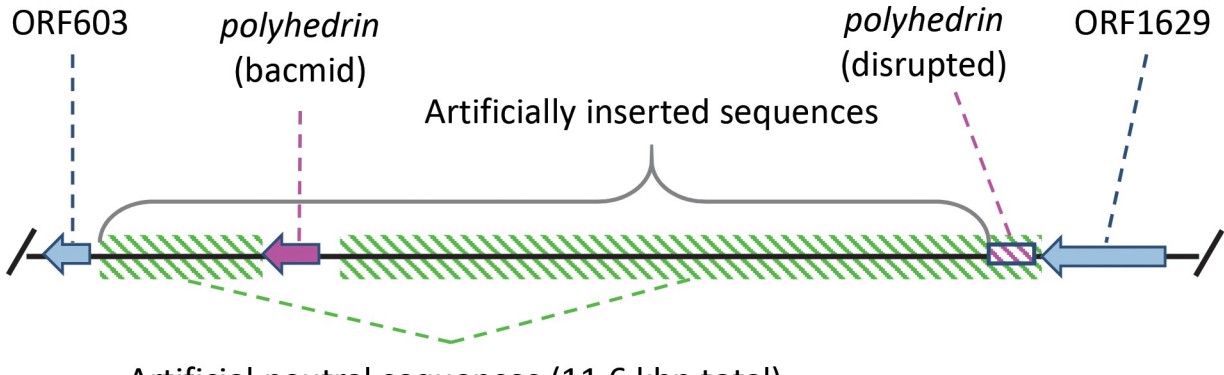

**Fig 1. Illustration of the neutral region (green diagonal bars) used for estimating mutation rate.** ORF603 and ORF1629 (light blue) are native AcMNPV genes, between which lies the *polyhedrin* gene which was disrupted (box with magenta bars) by the insertion of the bacmid sequences in its 5' end. To restore *polyhedrin* expression the gene has been reinserted under control of its native promotor within the bacmid insert. We consider the sequences of bacterial origin in the insert and the remnants of the pseudogenized *polyhedrin* gene copy as neutral sequences.

orally for the next passage. After five passages, each evolved population was amplified in 100 *S. exigua* L3 prior to further analyses. Further details on the experimental evolution experimental setup can be found in the Materials and Methods.

Evolved lineages A—E, as well as the ancestral BAC, were sequenced using Illumina HiSeq to detect mutations in both the non-functional genomic insert, as well as across the whole baculovirus genome. When mapping reads to the BAC reference genome we noticed a correlation between low sequencing coverage and the calling of mutations, and we therefore removed regions with relatively low coverage (S1 and S2 Figs, see Materials and Methods). We show that mutations indeed accumulated across lineages and passages (Table 1). When analyzing

**Table 1. Mutations called per lineage, where $\tau$ is the threshold frequency for detecting mutations and $\psi$ is the maximum number of lineages in which a mutation could occur before being filtered.**

| Lineage | $\psi$ | Neutral region only | | | Virus genome | | |
|---|---|---|---|---|---|---|---|
| | | $\tau = 0.5\%$ | $\tau = 1.0\%$ | $\tau = 2.0\%$ | $\tau = 0.5\%$ | $\tau = 1.0\%$ | $\tau = 2.0\%$ |
| A | 1 | 2 | 0 | 0 | 18 | 7 | 1 |
| | 3 | 4 | 2 | 0 | 54 | 16 | 2 |
| | 5 | 9 | 5 | 2 | 71 | 25 | 3 |
| B | 1 | 2 | 0 | 0 | 12 | 3 | 2 |
| | 3 | 4 | 0 | 0 | 19 | 3 | 2 |
| | 5 | 8 | 1 | 0 | 40 | 5 | 2 |
| C | 1 | 1 | 1 | 0 | 8 | 1 | 1 |
| | 3 | 2 | 1 | 0 | 35 | 7 | 1 |
| | 5 | 4 | 3 | 0 | 48 | 14 | 2 |
| D | 1 | 1 | 1 | 0 | 6 | 5 | 2 |
| | 3 | 3 | 1 | 0 | 8 | 5 | 2 |
| | 5 | 7 | 2 | 0 | 18 | 6 | 3 |
| E | 1 | 2 | 1 | 1 | 16 | 1 | 1 |
| | 3 | 4 | 1 | 1 | 47 | 8 | 1 |
| | 5 | 9 | 4 | 1 | 67 | 18 | 2 |
| Total | 1 | 8 | 3 | 1 | 60 | 17 | 7 |
| | 3 | 17 | 5 | 1 | 163 | 39 | 8 |
| | 5 | 37 | 15 | 3 | 244 | 68 | 12 |

the sequence data, the number of mutations detected is dependent on the minimum threshold value ($\tau$) for mutation frequency ($\tau$ values of 0.5, 1 and 2% were used for all analyses reported throughout this study). Mutations detected in the ancestral BAC were excluded from the analysis, as we were only interested in *de novo* mutations which occurred during the evolution experiment. We also noticed that some mutations were detected in multiple evolved lineages. We suspect that the majority of these observed mutations are due to sequencing and read-mapping errors, and consequently they should be removed from the set of mutations used for analyses. Nevertheless, to illustrate how this assumption affects the results, we have performed analyses with different values of the parameter $\psi$, the threshold for the number of evolved populations in which a mutation can occur. I.e., when $\psi = 1$ only unique mutations are included, and when $\psi = 5$ mutations detected in all five evolved populations are included. To limit the number of results presented, we report results for $\psi$ values of 1, 3 and 5 for most analyses. Regardless of the chosen mutation frequency threshold, the number of mutations detected is low for both the bacmid insert and genome after five passages in *S. exigua* L2, although the number of mutations detected increases as $\tau$ and $\rho$ increase (Table 1).

Mutations were not distributed randomly along the genome (S1 Fig): Kolmogorov-Smirnoff tests against a uniform distribution showed that mutation position is clustered, for both the bacmid region and the natural viral genome (Table A in S1 File). These analyses were performed for the lowest mutation frequency threshold ($\tau = 0.5\%$) to ensure sufficient mutations for a meaningful analyses. For the bacmid region and the high stringency condition for mutation calling ($\psi = 1$), the significance was only marginal (*P*-value = 0.062), whereas the results were significant to highly significant for all other tests. We expected mutations to be clustered, as the accuracy of the replication machinery is sequence dependent and consequently mutational hotspots have been described for many viruses [10].

## Low mutation rate for AcMNPV with bacmid and whole-genome mutation data

To make robust inferences on the mutation rate ($\mu$) from these experimental data, we developed a population genetic model. Briefly, we generated a stochastic model simulating neutral evolution in a virus genome, modelled as mutation rate per base per strand copying (s/n/r). We fitted this model to the experimental data by considering the number of bases with a frequency of mutations above the threshold $\tau$, using a maximum likelihood approach. We estimated the viral mutation rate to be $\mu \sim 1 \times 10^{-7}$ s/n/r, when filtering mutations that were detected in multiple evolved populations ($\psi = 1$). Whilst the threshold for mutation detection $\tau$ did not have a strong effect on mutation rate estimates, allowing mutations that occurred in multiple populations ($\psi > 1$) lead to higher mutation rate estimates (Fig 2). Estimates for the neutral bacmid region and the whole genome data gave similar results (Fig 2), although analyses of the rates at which different types of mutations accumulated suggests that mutations in the bacmid insert are indeed neutral. Finally, we explored how changes in viral demography–specifically the size of the population bottleneck and the final size of the population in each host–affect accumulation of neutral mutations, illustrating the importance of demography for mutation accumulation and hereby highlighting the importance of using a model for mutation rate estimation. Below we describe these results in more detail.

The simulation model was run for a range of model parameter values for mutation rate, viral replication mode ($\rho$, with values of 1, 3 and 10 used), threshold values for mutation detection ($\tau$ values of 0.5%, 1% and 2%), the maximum number of evolved populations in which mutations could occur ($\psi$ values of 1, 3 and 5), and lengths corresponding to the neutral bacmid region only and the whole virus genome (see Materials and Methods section for details

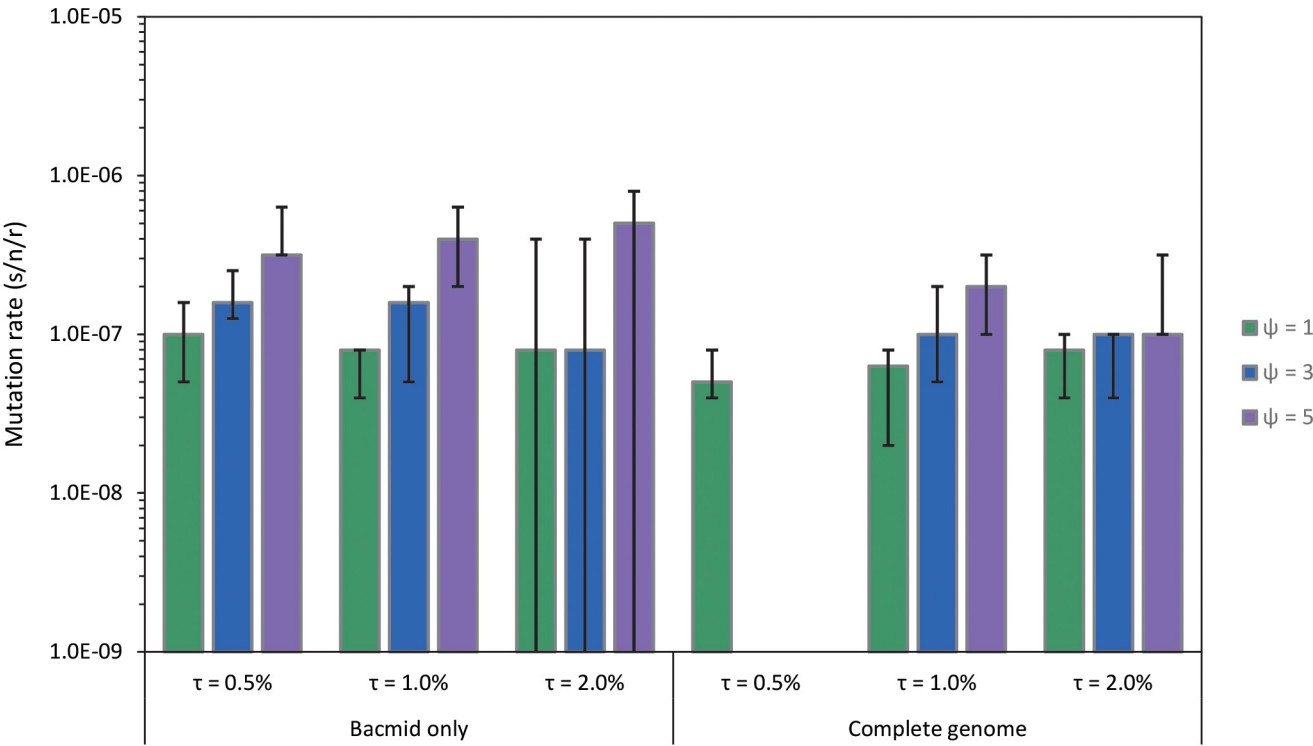

**Fig 2. Mutation rate estimates (s/n/r, mutations per site per strand copying) derived with the model are given based on the neutral bacmid region and the whole genome.** We estimated mutation rates for different values of the threshold frequency for detecting mutations ($\tau$), noted as percentages here, and for different values of the maximum number of lineages in which a mutation could occur before being excluded from the analysis ($\psi$). Error bars represent the 95% confidence interval, as determined by bootstrapping. Note that for the neutral region and $\tau = 2.0\%$, the lower fiducial limit extends to zero due to the low number of mutations detected.

and a full explanation of all parameters). Strikingly, our mutation rate estimates largely were robust relative to the replication mode $\rho$ value and to the choice of experimental data used (neutral region or whole genome), with estimates of approximately $\mu = 1 \times 10^{-7}$ s/n/r in all scenarios when $\psi = 1$ (Figs 2 and S3). When we included mutations that occurred in multiple populations ($\psi > 1$), in the most extreme case ($\tau = 0.5$, $\psi = 5$ and mutations in the bacmid region only) the estimated mutation rate was $\mu = 5 \times 10^{-7}$ s/n/r. As we think the repeated mutations are most likely sequencing errors, we think the estimate $\mu = 1 \times 10^{-7}$ s/n/r is the most valid.

We also estimated the mutation rate with established models for sequencing data from clones [9], adapting these methods to use the frequency of mutations determined by high-throughput sequencing data instead of Sanger sequences from clones (see Materials and Methods). These estimates were roughly similar to those obtained with the first approach, although they tended to be about a factor 2.5 smaller ($\mu \leq 4 \times 10^{-8}$ s/n/r when $\psi = 1$, S4 Fig). This difference was expected, as this alternative approach does not take into consideration the effects of the mutation detection threshold $\tau$, but rather assumes all mutations will be detected irrespective of their frequency. As the limited sensitivity of the deep-sequencing data is not taken into account, this approach likely will underestimate the mutation rate. Mutation rate estimates with established models also were higher if less stringent criteria for mutation calling were used, approaching $1 \times 10^{-7}$ when $\psi = 5$.

One of the purported strengths of the approach used here is the large region (11.6 kb) in which point mutations will not affect fitness, given there are no known viral genes or

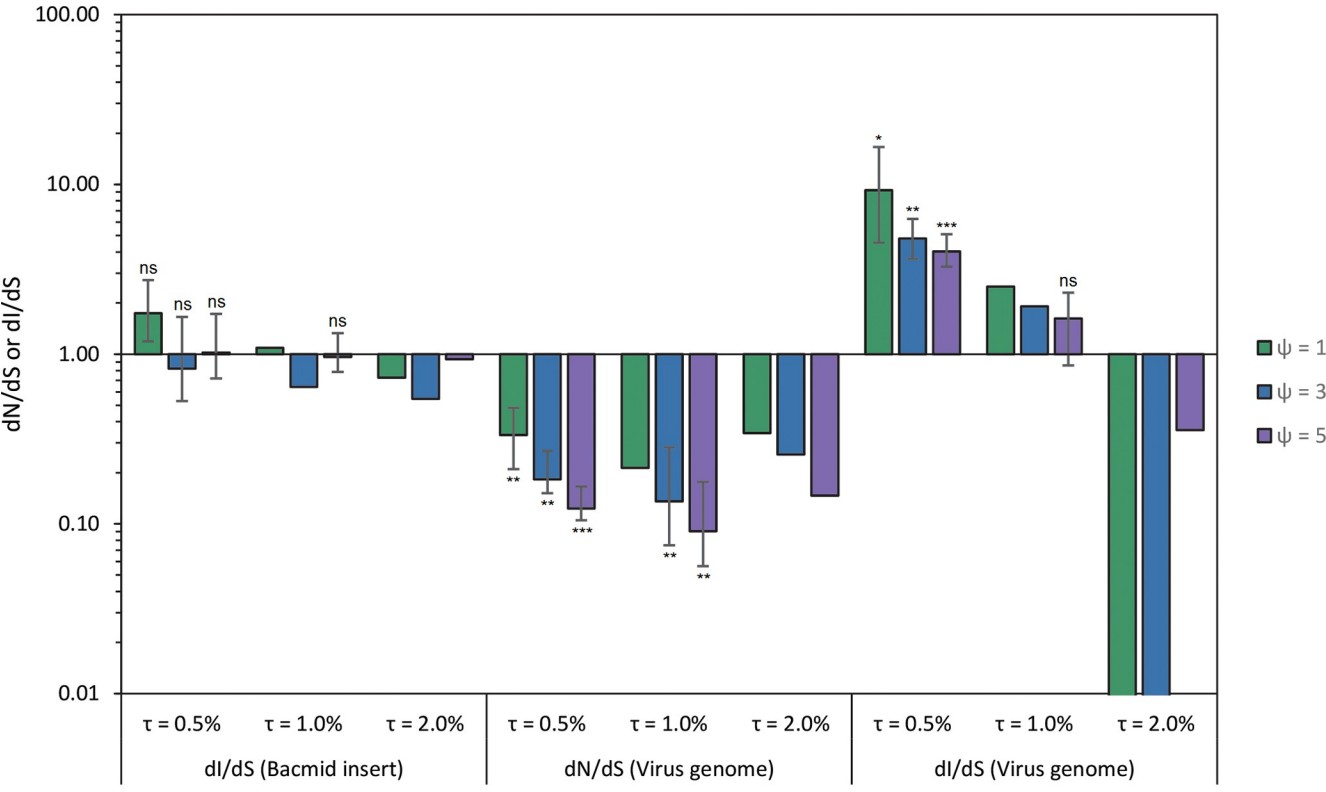

**Fig 3. Trends for the rates of intergenic (dI) or non-synonymous (dN) mutations are reported, all normalized by the rate of synonymous mutations in native viral genes (dS), for the different values used for the mutation threshold value (τ) and the maximum number of lineages in which a mutation can occur before being excluded from the analysis (ψ).** Error bars indicate the 95% confidence interval of the estimate as determined by bootstrapping, and results of a one-sample *t*-test on $\log_{10}$-transformed dI/dS or dN/dS values are indicated by ns (non-significant, $p > 0.05$), * ($p < 0.05$), ** ($p < 0.01$) and *** ($P < 0.001$). Note that for many conditions the confidence interval could not be determined, or the test could not be performed, as one or more values for a mutation class were zero, in which case no test results are indicated. The bars on the left labelled "dI/dS (Bacmid insert)" indicate the rate of intergenic mutations in the artificial neutral regions. These mutations occur with a normalized rate close to 1, indicating neutral evolution. The central bars labelled "dN/dS (Virus genome)" indicate the rate of non-synonymous mutations. These mutations were under-represented compared to synonymous mutations, indicating purifying selection. Finally, the columns on the right labelled "dI/dS (Virus genome)" indicate the rate of intergenic mutations in the viral genome, with bars extending to a value of 0.01 indicative of a value of zero. These results were inconclusive, as dI/dS was strongly dependent on the threshold for mutation detection chosen.

regulatory sequences (Fig 1). Although large genomic deletions in this region would presumably be beneficial as they could speed up viral replication [29], none were detected: Sequence coverage was similar to the rest of the genome for the evolved populations (S1 and S2 Figs), ruling out the occurrence of large deletions at high frequencies. To test if point mutations in this region were indeed neutral, we considered the rate of intergenic mutations (dI), normalized by the rate of synonymous substitutions in viral genes (dS) (see Materials and Methods). If mutations in the inserted region are neutral, we expect dI/dS ~ 1. For the different values of τ and ψ used, dI/dS estimates were indeed approximately 1 for the bacmid region, ranging from 0.54 to 1.75 (Fig 3). For all conditions for which we could perform a formal test, dI/dS was not significantly different from 1, lending further support to the idea that mutations in the bacmid region are neutral (Fig 3). By contrast, when the same analysis was performed for the rate of nonsynonymous mutations (dN) in viral genes, all dN/dS values were ≤ 0.34 (Fig 3). For all conditions under which we could perform a formal test, dN/dS was significantly lower than 1 (Fig 3). This underrepresentation of nonsynonymous mutations in the viral genome presumably occurs because most nonsynonymous mutations will be deleterious [13, 14], and

therefore are removed by purifying selection. Despite evidence for purifying selection acting on viral genes, mutation rate estimates were similar for the neutral region and the whole genome (Fig 2). Finally, we considered the normalized rate of intergenic mutations (dI/dS) for the authentic viral genome, and found a broad range of values ranging from 0 to 9.26. For the lowest value used for the threshold value for mutation detection ($\tau$ = 0.5%), intergenic mutations in virus genome were overrepresented; for a highest value ($\tau$ = 2.0%), there were few or no intergenic mutations (Fig 3). We therefore cannot draw any conclusions from these data, but note that almost all of the mutations found are associated with homologous regions (hrs). These repetitive elements will likely lead to sequencing and read-mapping errors, but their sequences are highly variable [30] and hence they could also be mutation hotspots.

The accumulation of mutations in a virus population is affected by the viral replication mode ($\rho$) [9, 10, 31–33], and the distribution of mutation frequencies is linked to the mode of replication [33]. Here we considered this effect by fitting the genome evolution model with values $\rho$ = 1 (the "geometric growth" scenario), $\rho$ = 3 (mixed replication scenario) and $\rho$ = 10 ("stamping machine" scenario). We found that viral mode of replication had a consistent but small effect on the estimated mutation rate (see S1 Text), with higher values of $\rho$ corresponding to higher mutation rate estimates, as expected. The effect on model fit was minimal (S3 Fig), and we therefore cannot make inferences on the mode of replication from these data. This result is not surprising, given that the number of mutations we detected was small and that the model fitting only considers the number of sites with a mutation frequency above $\tau$, and not the frequency of individual mutations. For baculoviruses, the mode of replication has not been described formally, to the best of our knowledge. However, as these viruses probably employ rolling circle amplification [34], replication is likely to follow the "stamping machine" scenario and to be described best by high values of $\rho$.

We considered the importance of two viral demographic parameters, the sizes of the founding ($\lambda$) and final population ($\kappa$) in an insect, and found that both parameters also had an effect on the accumulation of detectable mutations predicted by our model (S1 Text). This analysis showed a non-monotonic relationship between the size of $\lambda$ and accumulation of detectable mutations, with the highest numbers of mutations being detected at intermediate values of $\lambda$ (S5 Fig). Large values of $\lambda$ allow more mutations to be maintained in the virus population, but also prevent neutral mutations from reaching frequencies at which they can be detected (S6 Fig). This unintuitive result emphasizes the importance of considering demography when estimating viral mutations rates.

## AcMNPV mutation rate estimate is congruent with estimates for other viruses

Mutation rates (s/n/r) have been estimated for a number of viruses, allowing for a comparison with our baculovirus estimate. For this comparison, we included a collection of mutation rate data [9, 35], with updated mutation rates for the RNA viruses influenza A virus [12] and poliovirus [33] due to the availability of better estimates. When multiple mutation rates were available for one virus, we used only the most recent estimate because methodological advances make estimates that are more recent more reliable. Our estimate clearly is congruent with mutation rate estimates for other dsDNA viruses, as it is close to the predicted relationship between genome size and mutation rate (Fig 4). As AcMNPV has a relatively large genome, it is also one of the lowest estimates of mutation rate in DNA viruses reported in the literature, similar to that of Escherichia virus $\lambda$, and with only Enterobacteria phage T2 (170 kbp) being lower.

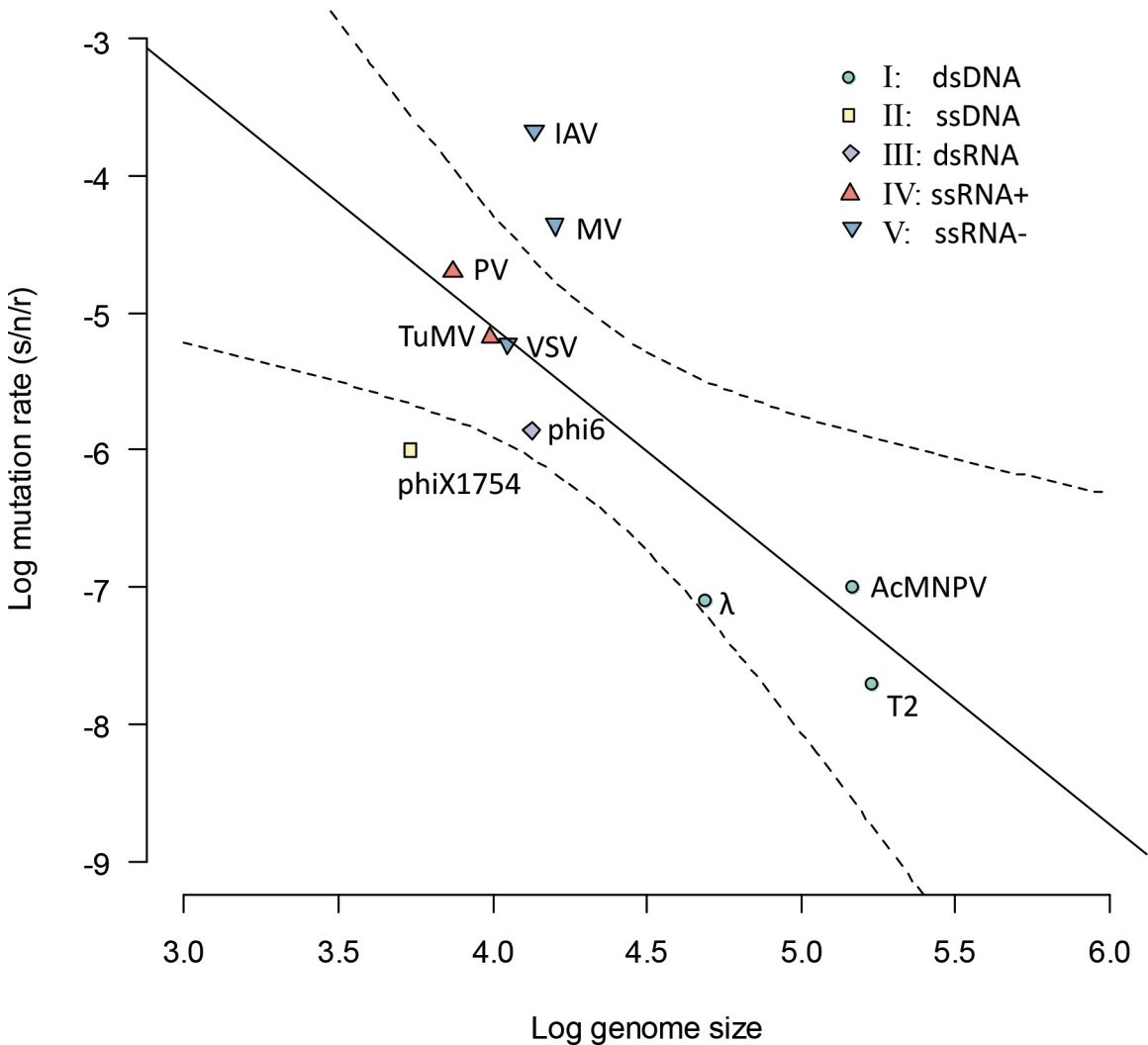

**Fig 4. An overview of known mutation rate estimates (s/n/r, ordinate) for viruses with different genome sizes (abscissa) is given.** The color and shapes of symbols indicate the Baltimore classification group to which each virus belongs, as indicated by the legend in the top right of the figure. The solid black line marks the regression line, with dotted lines marking the 95% confidence interval. The slope of the fitted relationship is significantly lower than zero ($t_8$ = -3.243, $p$ = 0.012), and the coefficient of determination ($r^2$) is 0.568. Our mutation rate estimate for AcMNPV is in good agreement with the fitted relationship between genome size and mutation rate. Other virus names in the figure are Enterobacteria phage T2 (T2), Escherichia virus λ (λ), Escherichia virus ΦX174 (phiX174), Influenza A virus (IAH), Measles virus (MV), Poliovirus (PV), Pseudomonas virus Φ6 (phi6), Turnip mosaic virus (TuMV), and Vesicular stomatitis virus (VSV).

Besides being consistent with trends in other viruses, our estimate of mutation rate for AcMNPV is congruent with what is known about its polymerase. AcMNPV codes for a DNA-dependent DNA polymerase (DNApol) that belongs to the family B DNA polymerases and contains the exonuclease domain, thought to be responsible for proofreading and editing of mismatches [16, 36, 37]. The 3' to 5' exonuclease activity of this domain—essential for repairing errors—has been confirmed [38], and hence a low mutation rate is expected for AcMNPV. The other viruses with low mutation rates are both dsDNA bacteriophages with relatively large genomes (Fig 4). Proofreading activity also has been demonstrated in the case of T2 [39].

### Analysis of the mutational spectrum: A low transition to transversion ratio?

We considered whether our data shed light on AcMNPV's mutation spectrum, the occurrence of the different kinds of single-nucleotide mutations, focusing on the transition to transversion ratio (Table 2; see also Tables B-D in S1 File). For the neutral region, the number of observed mutations is too small to be informative, even for the lowest mutation detection threshold of $\tau$ = 0.5%. We also considered mutation bias for the whole genome with different threshold values for mutation frequency ($\tau$) and the maximum number of lineages in which a mutation could occur ($\psi$). Whilst $\psi$ did not have a major effect on the transition to transversion ratio, this ratio depended strongly on $\tau$ (Table 2). For low values of $\tau$ the transition to transversion ratio was less than 1. For high values of $\tau$, the transition to transversion ratio becomes greater than one, but the number of mutations included in the analysis becomes very small and confidence interval for estimates becomes very large (Table 2). Moreover, for analysis outside of the neutral bacmid region selection may bias the mutations detected. Due to the small number of mutations detected in most conditions considered and the possible effects of selection on the whole-genome data, we therefore cannot draw any firm conclusions on AcMNPV's mutational spectrum. Analysis of a larger number of neutral mutational events will be necessary to draw conclusions on mutation spectrum. These larger numbers could be achieved by analyzing a larger number of replicate populations or populations evolved over a longer period of time, although improved sequencing methods with much lower error rates [40] may suffice to analyze mutation bias for the evolved populations described here.

### Concluding remarks

Our low mutation rate estimate is congruent with the large genome size of AcMNPV and its known proofreading activity [39]. By contrast, the high genetic variation often observed within alphabaculovirus populations [21–24] remains a conundrum. As baculovirus populations are subject to narrow bottlenecks at the start of infection [23, 41, 42], standing genetic variation will be rapidly lost [43] and a mechanism is required that introduces or maintains genetic variation. In one case, frequency-dependent selection has been observed in a baculovirus population and may account for stable polymorphisms [44, 45]. Whether intricate relationships between genetic variants that complement each other are common or evolutionarily stable remains to be seen, but such relationships would help to explain the genetic diversity often seen within alphabaculovirus populations. Recently, a novel strain of Chyrosodeixis includens nucleopolyhedrovirus was described, which generates high levels of genetic variation when it

**Table 2. Overview of the mutation spectrum, for different sets of mutations called.** Mutations were called with different values of the threshold frequency for detecting mutations ($\tau$), noted as percentages here, and for different values of the maximum number of lineages in which a mutation could occur before being excluded from the analysis ($\psi$). The number of transitions (transit.) and transversions (transver.) observed is noted, and below the transition to transversion ratio and its confidence interval (CI) are given.

| | | Relative frequency of mutation | | | |
| | | Whole genome | | | Bacmid only |
| $\psi$ | Observation or index | $\tau = 0.5$ | $\tau = 1.0$ | $\tau = 2.0$ | $\tau = 0.5$ |
| 1 | Transit. / Transver. | 27/41 | 9/11 | 6/2 | 3/5 |
| | Ratio [95% CI] | 0.659 [0.976–3.975] | 0.818 [0.460–3.329] | 3.000 [0.536–30.25] | 0.600 [0.093–3.082] |
| 3 | Transit. / Transver. | 78/102 | 13/31 | 6/3 | 7/10 |
| | Ratio [95% CI] | 0.765 [0.563–1.037] | 0.419 [0.202–0.825] | 2.000 [0.427–12.33] | 0.700 [0.225–2.040] |
| 5 | Transit. / Transver. | 129/152 | 29/54 | 10/5 | 19/18 |
| | Ratio [95% CI] | 0.849 [0.667–1.079] | 0.537 [0.330–0.859] | 2.000 [0.621–7.475] | 1.056 [0.524–2.135] |

is at low frequency in mixed infection [46]. If such strains with high mutation rates occur in other baculoviruses, their existence may help explain the high genetic variation often observed in baculovirus populations, even if they are rare genotypes. Our experimental setup uses a well-defined bacmid-derived virus population, and our mutation estimates will therefore be representative of a standard virus and not a mutator variant. Recombination could also help explain the high levels of genetic variation in natural baculovirus populations, as it could generate new haplotypes and enable mutations generated by mutators to spread widely through the population. Experimental work strongly suggests that baculoviruses have a high recombination rate [47].

We estimated the mutation rate for the alphabaculovirus AcMNPV, using an approach that depends on the insertion of a large artificial neutral region in the viral genome, and which starts with a single genotype and exploits the genome stability of group I dsDNA viruses. Such an approach requires mutations in this 'artificial' region to be neutral, and dI/dS results suggest this assumption is met. We developed an approach to analyzing regular Illumina sequencing data that were not gathered specifically with the intent of determining mutation rates. Our approach relies on a detailed analysis of the sequencing data to eliminate obvious sequencing biases and a comparison of sequencing data to simulation-model predictions. The use of models allows us to take into consideration the effects of viral demography on mutation rate estimates, and limit the impact of choosing thresholds for mutation detection, as the same threshold is applied in the model. Others have employed high fidelity approaches such as duplex sequencing to reduce sequencing error associated with high-throughput sequencing [40] for estimating mutation rates of DNA viruses and finding mutation-prone sequence motifs and mutational hotspots [27]. High throughput sequencing has also enabled the characterization of genetic diversity within DNA virus populations and its functional implications [48, 49]. Long-read sequencing has made it possible to study of evolutionary dynamics of adaptation by point mutations and gene-copy-number variation in poxviruses [50]. In the future, combining this sequencing technology with large neutral regions could make it viable to extend work on rates and the distribution along the genome of mutational events beyond point mutations to include structural mutations, such as large indels or copy number variation [50, 51].

We estimated the mutation rate per base per strand copying (s/n/r), as has been done in many other previous studies [9, 10]. This metric is convenient, as we could make empirically supported estimates of the initial and final viral population sizes in infected larvae (see Materials and Methods). By contrast, the mode of replication for baculoviruses has not been quantified, and we therefore considered the effect of this parameter, but did not find strong effects on model fit or the mutation rate. An alternative approach would have been to calculate the mutation rate per base per cell infection (s/n/c) [9, 10], in which case we would be missing many details of cellular infection dynamics, particularly in the early stages of infection.

Some mutations were detected in multiple evolved populations, whilst not being detected in the ancestral bacmid. Mutations that were detected in the ancestral bacmid simply could be excluded from analyses because they did not occur *de novo* during the experiment, or alternatively were indicative of sequencing or read-mapping errors. For the repeated mutations that were not detected in the bacmid, the most parsimonious explanation is that they are sequencing or read-mapping errors. Even with strong selection and a large effective population size, it is unlikely that multiple mutations will be presented in all evolved populations [51, 52]. Moreover, our dI/dS and dN/dS results suggest neutral molecular evolution predominates the bacmid region, and purifying selection predominates in the viral genome. Hence, these changes would need to be driven exclusively by mutation bias, an explanation that we find unlikely. However, since we cannot categorically rule out this explanation, for all our data analysis we

considered the implications of including repeated mutations. In the most extreme case this lead to a mutation rate estimate of $\mu = 5\times10^{-7}$ s/n/r, half an order of magnitude higher than what we consider the best estimate of $\mu = 1\times10^{-7}$ s/n/r when repeated mutations are excluded (see Fig 3). Although the criteria for including mutations affect the mutation rate estimate, even when we apply less stringent conditions this estimate will still be relatively low.

We chose to perform our experiments in *S. exigua*, primarily because in our experience we could perform experiments in this particular population of hosts–including reconstitution of the virus from the infectious clone–without activation of latent viruses. Although we used the bacmid region to estimate mutation rates, evolutionary dynamics in the natural viral genome might still have an effect on the observed mutations in the bacmid region. For example, if neutral mutations in the bacmid region hitchhike to high frequencies with a beneficial mutation in the viral genome, this could effect the final distribution of mutation frequencies observed in the bacmid. Although *S. exigua* is sometimes considered a semi-permissive host for AcMNPV the particular *S. exigua* colony used for our experiments is highly susceptible to AcMNPV (E. g., the infectivity of OBs to early instar larvae is similar to that in *Trichoplusia ni*, a permissive host [41]), suggesting the scope for adaptation may be limited. Indeed, we found few mutations in the viral genome for the evolved lineages, and an analysis of the rate of nonsynonymous and synonymous substitutions suggests purifying selection predominated. These observations make it unlikely that positive selection on beneficial mutations in the viral genome distorted the observed distribution mutations in the bacmid region or affected mutation rate estimates.

## Materials and methods

### Experimental evolution of AcMNPV

pBac-E2 (BAC) [28] was evolved experimentally in *S. exigua* larvae in five replicate lineages (A, B, C, D and E). To reconstitute the virus from the bacmid, the haemocoel of *S. exigua* L4 was injected with a total volume of 20 μl (i.e., $2 \times 10$ μl), containing a 4:1:1 mixture of Lipofectin transfection reagent (ThermoFisher Scientific), water and BAC DNA (~ 15 μg DNA per larva). Upon larval death, OBs were harvested from cadavers. From a single infected larva, OBs were isolated and diluted to $2 \times 10^{7}$ OBs/ml. Serial passage of BAC was performed five times for each replicate lineage, with five larvae used for each replicate lineage. For each passage, newly molted L2 were starved for 12 h and then inoculated by droplet feeding with an OB suspension exceeding 10 x $LC_{99}$ ($\geq 2 \times 10^{7}$ OBs/ml), to avoid narrow transmission bottlenecks. Per replicate lineage, five inoculated larvae were transferred to 6-well tissue culture plates with artificial diet plugs. Larvae were incubated at 26˚C and with a 14 h:10 h day-night photoperiod. Upon death, larval cadavers were collected, pooled and used to inoculate the next passage. After five passages, lineages A—E were amplified using 100 *S. exigua* L3 exposed to a high concentration of OBs ($3 \times 10^{9}$ Obs/ml) by droplet feeding, and $1.5 \times 10^{9}$ OBs were used to extract viral genomic DNA. Briefly OBs were dissolved with DAS buffer (0.1 M $Na_2CO_3$, 0.15 M NaCl, 10 mM EDTA, pH 11), and DNA was then extracted from the liberated occlusion-derived virus particles using a DNA isolation kit (Omega Bio-tek) following the manufacturer's instructions. For the BAC, DNA was extracted from 50 ml LB from an overnight culture using a plasmid midi kit (Qiagen). Successful genomic DNA extraction was confirmed by PCR with primers gp41 inner F (5'-CAAGAGCAAAGAACCGACG-3') and inner R (5'-TTATG-CAGTGCGCCCTTTCGT-3'), and contamination of SeMNPV was ruled out by PCR with primers Se F (5'-GACGACGAATTATGTTGTGACCGAC-3') and R (5'- AGATGGATG-GAAAGGCAACGCT-3'). Purified DNA (~ 1.5 ug) from the evolved AcMNPV lineages A, B, C, D and E, as well as the ancestral BAC, was used for library preparation with the Next Ultra

DNA Library Prep Kit for Illumina (New England Biolabs), followed by Illumina HiSeq paired-end 150 (PE150) sequencing (Beijing Novogene Bioinformatics Technology Co., Ltd). The raw sequencing data are available in the Sequence Read Archive under accession PRJNA798700 (https://www.ncbi.nlm.nih.gov/sra/PRJNA798700).

## Mutation calling and filtering

Because the number of viral reads was not equal across samples, fastq files were subsampled to ensure an approximately equal mean coverage across the reference genome for each isolate using seqtk sample [53]. NGS data was analyzed using CLC Genomics Workbench 20.0 [54]. Reads were trimmed (quality limit = 0.05) and mapped to a reference genome. The reference genome is based on the sequence of the E2 variant [55], the details of the original bacmid construction and donor vectors [28], and limited Sanger sequencing to bridge small gaps (S2 File). Mutations were called using the "low frequency variant detection tool" (minimum frequency = 0.5%). An overview of parameter settings is included (S1 Data). Additional filtering criteria were: forward-reverse balance > 0.05, read count > 10, and the type of mutation is "SNV" (single nucleotide variant). Moreover, positions with extreme coverage values were excluded. To this end, we ranked the coverage value per position for each lineage and excluded the upper and lower 1%. Analyses were done with three thresholds $\tau$ for mutation frequency: 0.5%, 1% and 2%, and thee threshold values $\psi$ for the number of lineages in which the exact same mutation could occur before it was filtered: 1, 3 and 5 (S2 and S4 Data, respectively). Finally, mutations were tallied per isolate, both across the whole genome and for the neutral bacmid insert (S3 File).

## Mutation model and mutation rate estimation

We generated a stochastic model that predicts the distribution of mutation frequencies per base in an evolving virus genome, and then fitted this model to our empirical data with a maximum likelihood approach to obtain mutation rate estimates. The model was implemented in R 4.0.3 [56] and all code is available (S1 and S2 Code). We model the genome region under consideration as a vector with $g$ elements for each nucleotide position, with each element representing the total frequency $f$ of mutated bases at position $i$. For simplicity, we do not consider the identity of the mutated bases and we do not allow for reversions, as we are considering scenarios in which mutations are rare and the probability of a reversion occurring and reaching high frequency is very low. We assume that all mutations are strictly neutral, and that all changes in mutation frequency result from the occurrence of *de novo* mutations or neutral processes like stochastic changes in allele frequencies due to population bottlenecks (i.e. genetic drift). Parameter values, additional explanation and justification are provided for the fitted (Table 3) and fixed (Table 4) model parameters.

At the start of the infection of an individual host, there is a bottleneck with $\lambda$ virus genomes initiating infection. We first draw the number of occlusion derived viruses that infect the host, allowing it to follow a zero-truncated Poisson distribution with a mean of $\lambda/\zeta$, where $\zeta$ is the

**Table 3. Fitted parameters for the models for estimating mutation rates.**

| Parameter | Value | Explanation |
|---|---|---|
| $\rho$ | 1, 3, 10 | Parameter that describes the viral mode of replication, with 1 being equivalent to equivalent to "geometric growth" by fission and large values (i.e., 10) representing "stamping machine" replication kinetics. |
| $\log_{10}\mu$ | -10, -9.9, -9.8, [. . .], -5 | Mutation rate (s/n/r) |

**Table 4. Fixed parameters for the models for estimating mutation rates.**

| Parameter | Value | Explanation |
|---|---|---|
| $g$ | 11,646 | The length of the neutral bacmid region in bases. |
| | 145,465 | The length of the full genome, including the neutral bacmid region. |
| $\lambda$ | 46 | The bottleneck size for the viral founding population in each insect larvae. Following [41], the bottleneck size is related to the host survival ($S$) for AcMNPV infection of *S. exigua*: $\lambda = \ln(S)$. For each insect exposed to 10 x $LD_{99}$ dose: $\lambda = 10 \times -\ln(1-0.99) \sim 46$. As is typical for multicellular host/virus pathosystems, a virus inoculum with a large number of horizontal transmission stages is used, but the ensuing bottleneck is still narrow [23, 42]. |
| $\zeta$ | 3.71 | Single parameter for the zero-truncated Poisson distribution of nucleocapsids per occlusion derived virus (ODV) [57]. Note that the corresponding mean of the distribution is 3.80 nucleocapsids per ODV. |
| $\kappa$ | $5.05 \times 10^8$ | The final size of the virus population within a single L2. As AcMNPV generates OBs with multiple ODVs, and ODVs with multiple nucleocapids, each containing single copy of the genome. The mean OB yield per larvae during the experiment was $1.33 \times 10^6$, based on the OB concentrations measured for each pool of five insects made after each round of passaging, assuming 100 ODV per OB (we are not aware of any empirical estimates) and a mean of 3.8 nucleocapsids per ODV [57]. |
| $\sigma$ | 1001 | Parameter value minus one indicates the maximum number of mutations per genome allowed in the simulations. |
| $\varphi$ | $10^4$ | Threshold value of population growth for switching from stochastic to deterministic mutation. |
| $\tau$ | 0.5%, 1%, 2% | Minimum frequency for the detection of mutations, indicated as a percentage. |
| $\psi$ | 1,2, [. . .], 5 | The threshold value for the number of evolved lineages in which a mutation can occur for the empirical data. I.e., the most stringent condition is $\psi = 1$, as only mutations that occur in one lineage are accepted. When $\psi = 5$, mutations that occur in all 5 lineages are accepted. |

number of nucleocapsids (each containing one genome copy) per occlusion derived virus (ODV). We obtained an estimate of $\lambda = 46$ by considering the relationship between host mortality and the number of viral founders [41] (see Table 4). We use a zero-truncated distribution to avoid having uninfected hosts, but this approximation does not affect our results as the dose used in experiments was high (10 x $LD_{99}$ dose) and virtually no hosts remain uninfected. Next, for each infecting ODV, we draw the number of nucleocapsids contained from a zero-truncated Poisson distribution with a mean $\zeta$, as the multiple nucleopolyhedroviruses have multiple nucleocapsids present in each ODV. Our model therefore incorporates stochasticity in the number of infecting virus particles and their nucleocapsid content. For each position in the genome, we draw the number of genomes containing a mutation at this position following the population bottleneck at the start of infection from a binomial distribution, where for the $i^{th}$ position: $P(X_i = x_i) = \binom{\lambda}{x_i} f_i^{x_i}(1 - f_i)^{\lambda - x_i}$, where $x$ is the number of mutant genomes added to the population at a particular step in the infection process (and $f$ is the frequency of mutated bases at position $i$). The virus population then expands within the host exponentially with a replication factor $\rho$ per cycle of virus replication within the host, such that $N_t = \lambda(1+\rho)^t$ where $N$ is the number of genomes present at a time $t$, measured in generations of viral replication within the host. It is unknown what the mode of replication [31, 32] is for a baculovirus. We therefore used values of 1 ("geometric" or "symmetric" replication with a doubling of the number of copies per cycle–one original genome copy and one replicated copy), 3 ("mixed" replication–one original genome copy and three replicated copies) and 10 ("stamping machine" or "asymmetric" replication) for $\rho$. Replication proceeds until the carrying capacity $\kappa$ of a host is

reached, with an expansion to exactly $\kappa$ virus genomes allowed in the final round of replication. During each round of replication, the number of new mutants that occur at each position follows a binomial distribution, such that the mutation rate $\mu$ is the probability of success and $\eta$, the number of genomes generated during that round of replication which are not mutated at this nucleotide position (i.e., $\eta_{i,t} = \rho N_{t-1}(1 - f_{i,t-1})$), is the number of events:

$P(X_{i,t} = x_{i,t}) = \binom{\eta_{i,t}}{x_{i,t}} \mu^{x_{i,t}}(1 - \mu)^{\eta_{i,t} - x_{i,t}}$. However, to make the model computationally tracta-

ble for large population sizes, we switched to deterministic mutation (I.e., $X_{i,t} = \eta_{i,t}\mu$) once a large number of virus genomes was being generated relative to the mutation rate ($N_t \varphi > \mu^{-1}$), where $\varphi$ is a constant. Note that all mutations are assumed to be neutral and that the bottleneck at the start of infection is narrow ($\lambda = 46$). Mutations occurring late in infection when the viral census population size is large therefore will rarely be sampled during the bottleneck events at the start of the next round of infection (i.e. serial transfer). To model the infection of five host larvae per replicate, we simulated five separate infections and pooled the viral progeny by taking the mean mutation frequency per site over the five larvae for each position in the genome, and using these $f_i$ values for the next round of infection. After 5 rounds of infection, we also included the final amplification of the virus in 100 larvae to generate the final population that is compared to the sequencing results. For L3, we assumed the same population bottleneck $\lambda$ as for L2 –as a high OB concentration was used for final amplification– and the number of viral genomes generated was double that for L2 ($2\kappa$).

To fit the model to the data, we first ran 1000 simulations for each combination of parameter values (i.e., $\mu$, $\rho$ and $\tau$) to generate model predictions. We compared the observed number of experimental replicates ($q_j$) with $j - 1$ mutated nucleotide positions with a frequency $f_i$ higher than the threshold value $\tau$, to the frequency predicted by the model ($\beta_j$). (I.e., $q_1$ is the number of experimental replicates with 0 nucleotide positions for which $f_i > \tau$, $q_2$ is the number of replicates with 1 nucleotide positions for which $f_i > \tau$, etc.) The multinomial pseudo-likelihood of any realization is then: $P(Q_1 = q_1, Q_2 = q_2, [\ldots], Q_\sigma = q_\sigma) = \frac{q!}{\prod_j^\sigma q_j!} \prod_j^\sigma \beta_j^{q_j}$, where $\sigma - 1 = 1000$

is the maximum number of mutated bases that were tracked in the simulations. If the number of mutated nucleotide positions (with a frequency $f_i$ higher than the threshold value $\tau$ for detection) exceeded $\sigma - 1$ in any simulation, results from that set of parameter values were excluded from further analysis. We fitted the model to 1000 bootstrapped datasets to determine the 95% confidence interval of the parameter estimates.

## Mutation rate estimates with established approaches

We estimated the mutation rate with an established approach, to compare to the estimates made with our simulation-model approach. A canonical approach [9] for calculating mutation rate (s/n/r) is $\mu = \frac{3m_s}{T_s c \alpha}$, where $m_s$ is the number of observed mutations observed in sequenced clones, $T_s$ is the mutational target size, $c$ is the number of viral generations (i.e., in terms of strand copying events), and $\alpha$ is a correction for the effects of selection. To obtain $m_s$ from our deep sequencing data, we sum the frequency of all observed mutations ($f_i$) above the threshold for mutation selection $\tau$ in all lineages. If these sequencing data are accurate and mutations are neutral, the frequency of each mutation is also the probability that this mutation would be detected in a randomly selected clone by sequencing. This approach is a simple approximation, as here we do not consider the effect of the threshold for mutation detection ($\tau$) on mutation rate estimates. (Lowering $\tau$ will lead to a larger number of mutations and consequently a higher mutation rate estimate. To keep this method as simple as possible and free of additional assumptions, we choose not to incorporate any corrections.) Note that because we sum the

frequency of all possible mutations over each site, we can drop the three in the numerator. To obtain $T_s$, we multiply the length of the neutral region by the number of replicates. To obtain $c$ we estimate the number of generations assuming different values for $\rho$ (i.e., 1, 3 and 10 as for the simulation-based model fitting), such that $c = \theta \cdot ln(\kappa/\lambda)/ln(1+\rho)$, where $\theta$ is the number of passages. Finally, we can drop $\alpha$ because we only consider mutations in the neutral bacmid region. One-thousand bootstrapped datasets were used to obtain fiducial limits for the mutation rate estimates. To make predictions of mutation accumulation ($m_s$) using this model (I.e., see the legend of S5 Fig), we use the simplified relationship $m_s = \mu T_s c$.

### dI/dS and dN/dS analyses

Estimates of dN/dS (i.e., the normalized rate of nonsynonymous mutations, here made for authentic viral genes) were made using standard methods [58, 59]. As we are not aware of any estimates of the mutation spectrum for insect DNA viruses and our own data suggest these biases may not be very strong (Table 2), we assume no mutation bias is present (i.e., a transition to transversion ratio of 1). For the dI/dS [3], the dS term is the same as for the dN/dS analysis, derived from the results for natural viral genes. The dI term is determined for mutations in the bacmid neutral region, or for the intergenic regions of the natural virus genome. Ninety-five percent confidence intervals of the dI/dS and dN/dS were obtained using 1000 bootstrapped datasets, and data were tested for significance with a one-sample $t$-test on the dN/dS values calculated for individual samples compared to a test value of 1. However, when $\tau$ > 0.5% for one or more samples the number of intergenic, non-synonymous or synonymous samples was 0, and hence these analyses could only be performed for $\tau$ = 0.5%. Full results and R code have been made available (S3 and S4 Code and S5 Data).

## Supporting information

**S1 Fig.** We show coverage along the genome for each evolved line (A, B, C, D and E) as well as ancestral strain BAC. Position of mutations observed at mutation frequency threshold value ($\tau$) = 0.5 and present only in a single evolved population ($\psi$ = 1) are shown as black dots. Coverage patterns are similar between the different isolates. The peak observed at around 10000 bp for the BAC isolate is due to the presence of empty bacmid vectors in sequencing data and is omitted from mutation calling.
(TIF)

**S2 Fig. Coverage distribution per isolate after subsampling to approximately equal mean coverage.** Isolates have a mean coverage of around 5500. The BAC isolate is showing an additional peak at a coverage of around 10000, which is explained by the presence of empty bacmid vectors in sequencing data.
(TIF)

**S3 Fig. We show the negative log likelihood (NLL) for models fitted with different mutation frequency threshold values ($\tau$), given as a percentage.** For simplicity, we show the results when only unique mutations are considered ($\psi$ = 1). Mutation frequency threshold values clearly effect model fit, as they have an effect on the number of mutations that will be detected. By contrast, assumptions on the value for the parameter that determines the mode of virus replication ($\rho$) had little effect on model fit. This result is not surprising however, given that our model does not consider the frequency of mutations, but simply the number of bases with a mutation frequency greater than $\tau$.
(TIF)

**S4 Fig. Estimate of mutation rate (s/n/r) using established methods applied to deep sequencing data for the bacmid region (solid bars, samples categorized as "Classic"), as an alternative to our approach using a simulation-based model (hatched bars on the right, samples categorized as "Simulation model").** Mutation rates were estimated for different values of the viral mode of replication ($\rho$), different values for the threshold of mutation detection ($\tau$), and different values for of the maximum number of lineages in which a mutation could occur before being excluded from the analysis ($\psi$). Error bars represent the 95% fiducial limits, as determined by bootstrapping. When the lower fiducial limit extends beyond the lower limit of the axis, this indicates a lower fiducial limit of zero. Overall, these estimates were lower than those obtained with the approach employing a simulation model. As baculoviruses most likely employ rolling circle amplification, replication is likely to have a high value of $\rho$. Therefore, the best estimates with this approach assume the highest value of $\rho$. Moreover, they will assume the lowest mutation detection threshold ($\tau$), provided all mutations are assumed to be bona fide, as the cumulative frequency of mutations above this value is used required to estimate mutations and no correction is made for this threshold. Finally, as in our other analyses, we think the most conservative estimate of mutation rate will exlude all repeated mutations ($\psi = 1$). These conditions ($\rho = 10$, $\tau = 0.5\%$, $\psi = 1$) render an estimate of $\mu = 3 \times 10^{-8}$ s/n/r, which is lower but roughly similar to for our simulation-based approach ($\mu \sim 10^{-7}$).
(TIF)

**S5 Fig. These heatmaps indicate the number of mutations that accumulate after 5 passages in 5 insects, based on the predictions from the simulation model.** For all simulations, we assumed a mutation rate similar to our estimated value for baculoviruses ($\mu = 10^{-7}$), and kept other model parameters the same as for model fitting (Table 4) unless otherwise indicated. We varied the size of the founding viral population in one insect ($\lambda$, x-axis is the $\log_{10}[\lambda]$) and the final size of the viral population in one insect ($\kappa$, y-axis is the $\log_{10}[\kappa]$), while also varying the threshold value for mutation detection ($\tau$) and mode of virus replication ($\rho$) over the different panels. The purple cross indicates the point in the parameter space that corresponds to the model parameters assumed in model fitting ($\lambda = 46$, $\kappa = 5.05 \times 10^8$). There are more detectable mutations when $\tau$ is low, when $\rho$ is low, and as the final population size $\kappa$ increases. Increases in the size of the founding viral population $\lambda$ initially lead to increasing numbers of detectable mutations, but the number of detectable mutations eventually decreases. For an explanation of this non-monotonic behaviour, see S6 Fig. Finally, note that we can also predict mutation accumulation using the established approach (see Materials and Methods Section) for comparison purposes, which does not take $\tau$ into account. The range of model predictions for the number of accumulating mutations (lowest to highest predicted value, based on the extreme values of $\lambda$ and $\kappa$) is then: for $\rho = 1$, 0.48–2.42 mutations; for $\rho = 3$, 0.24–1.21 mutations; for $\rho = 10$, 0.13–0.70 mutations. The simulation model which takes into consideration better the effects of demography on mutation accumulation, therefore predicts considerably lower and higher mutation accumulation under some conditions.
(TIF)

**S6 Fig. An illustration is provided of why the size of the founding population ($\lambda$) has a non-monotonic effect on the accumulation of detectable mutations.** The simulation model was run for 5 passages in single insect larvae, with a genome size of $g = 50,000$ base pairs, mutation rate $\mu = 10^{-7}$ and final population size $\kappa = 3 \times 10^8$. We then varied $\lambda$, as indicated at the top of each column of panels, with all panels in a column simply representing replicate simulations. We plotted of the $\log_{10}$-transformed frequency of mutations at each position (y-axis) at the end of each round of passaging (x-axis), randomly selecting a hue and line type for each position to make them easier to distinguish. Finally, for each panel we noted the number of

mutations which were above a frequency of 0.01 (*a*, with the threshold indicated by a blue line) and mutations above a frequency of 0.0001 (*b*, with the threshold indicated by a purple line). We assume the *a* mutations will be detected by sequencing, as $a \sim \tau$, the threshold value for mutation detection used. The *b* mutations are sometimes maintained in the population over passages, but they need not be detected as they can be below $\tau$. The number of *b* mutations increases as $\lambda$ is increased, whereas the number of *a* mutations only increases initially. Wide bottlenecks will lead to the maintenance of more mutations in the population, but they also limit the stochastic increases in mutation frequency and prevent mutations from reaching the detection threshold. Recall that all mutations are assumed to be strictly neutral, and that all changes in mutation in mutation frequency are due to *de novo* mutations or genetic drift. (TIF)

**S1 Code. R script for estimating mutation rates based on the bacmid region data.**
(TXT)

**S2 Code. R script for estimating mutation rates based on the whole genome data.**
(TXT)

**S3 Code. Main R script for the dN/dS analysis.**
(TXT)

**S4 Code. R script for bootstrapping and statistical tests for dN/dS analysis.**
(TXT)

**S1 Data. PDF file with CLC Genomics Workbench settings.**
(PDF)

**S2 Data. Notebook with scripts and results for mutation calling ($\tau$ = 0.5%).**
(HTML)

**S3 Data. Notebook with scripts and results for mutation calling ($\tau$ = 1%).**
(HTML)

**S4 Data. Notebook with scripts and results for mutation calling ($\tau$ = 2%).**
(HTML)

**S5 Data. Excel file containing the final results for the dN/dS analysis.**
(XLSX)

**S1 File.** PDF file with supplementary tables, including Table A (Analysis of the distribution of mutations along the genome), Table B (Relative frequencies of mutations, $\psi$ = 1), Table C (Relative frequencies of mutations, $\psi$ = 3), and Table D (Relative frequencies of mutations, $\psi$ = 5). (PDF)

**S2 File. ZIP file containing the bacmid sequence and annotation files used in the genome analysis here (sequence as $^*$.fa file, annotation as $^*$.csv and $^*$.gbk files).**
(ZIP)

**S3 File. ZIP file containing 36 $^*$.csv files, containing the mutations called for each condition (whole genome vs bacmid region only, threshold for repeated mutations, and threshold mutation detection) as indicated by the file names.**
(ZIP)

**S1 Text. PDF file containing Supplementary Text 1 (Relevance of viral demography for mutation rate estimates).**
(PDF)

## Acknowledgments

The authors thank Yue Han and Marleen Henkens for technical assistance.

## Author Contributions

**Conceptualization:** Dieke Boezen, Ghulam Ali, Wopke van der Werf, Just M. Vlak, Mark P. Zwart.

**Formal analysis:** Dieke Boezen, Mark P. Zwart.

**Investigation:** Ghulam Ali, Manli Wang, Xi Wang.

**Supervision:** Wopke van der Werf, Just M. Vlak, Mark P. Zwart.

**Writing – original draft:** Dieke Boezen, Just M. Vlak, Mark P. Zwart.

**Writing – review & editing:** Dieke Boezen, Ghulam Ali, Manli Wang, Xi Wang, Wopke van der Werf, Just M. Vlak, Mark P. Zwart.

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
