## [Decision Letter · Decision Letter 0]

19 Oct 2021

Dear Dr Zwart,

Thank you very much for submitting your Research Article entitled 'Empirical estimates of the mutation rate for an alphabaculovirus' to PLOS Genetics.

The manuscript was fully evaluated at the editorial level and by three independent expert peer reviewers. The reviewers appreciated the attention to an important problem, but you will see that they have varied opinions on the suitability of your paper for PLOS Genetics and the general strengths of your findings. In particular, concerns were raised about the MOI and passaging conditions that could challenge the estimates you have generated. We feel that you may be able to address these concerns with a detailed rebuttal and a revision which will be strengthened by incorporating some additional experiments based on the constructive suggestions made here. Based on the reviews, we will not be able to accept this version of the manuscript, but given the likely importance of this paper to baculovirologists and dsDNA virologists, we would be willing to review a much-revised version. We cannot, of course, promise publication at that time.

If you decide to revise the manuscript for further consideration at PLOS Genetics, please aim to resubmit within the next 60 days, unless it will take extra time to address the concerns of the reviewers, in which case we would appreciate an expected resubmission date by email to plosgenetics@plos.org.

[LINK]

We are sorry that we cannot be more positive about your manuscript at this stage. Please do not hesitate to contact us if you have any concerns or questions.

Yours sincerely,

Harmit S. Malik

Associate Editor

PLOS Genetics

Bret Payseur

Section Editor: Evolution

PLOS Genetics

Reviewer's Responses to Questions

**Comments to the Authors:**

Reviewer #1: PGENETICS-D-21-01205

This manuscript describes an estimate of the mutation rate of an alphabaculovirus, AcMNPV, by deep sequencing of a modified genome that comprises a DNA region that is supposed non-expressed in insect cells, related to the replication ability in bacteria (Bacmid).

The manuscript is well written and easy to read, and the methods are clearly described.

The authors analysed the mutations that accumulated in the genome on five parallel lineages after five passages through Spodoptera exigua larvae at high multiplicity of oral infection.

Three major hypothesis are taken in the manuscript : i.) The bacterial region is perfectly neutral, ii.) all progeny genomes have the same mutation rate, and iii) the mutation rate is the same all through the genome.

i). The authors argue that the low copy number bacterial insert can be considered as neutral, and provide evidence of this neutral character in Figure 3 by comparing the rate of synonymous versus non synonymous mutations in this regions compared to the genuine baculovirus genes. For the former, there is conclusion of neutrality, while for the later there is a global conclusion of purifying selection. As not all virus genes are selected at the same level, it could be of interest to differentiate them into “highly selected and loosely selected”. However, is the number of mutations enough for that approach to be carried out?

ii). The presence of a exonuclease activity in the viral DNA polymerase seems to be associated to the low mutation rate observed on baculoviruses. There is a recent work (Aguirre et al. Viruses 2021) showing that in an alphabaculovirus, one variant is able to generate high levels of variation. This is not the first time that high variation is observed in baculovirus populations. Although this observations do not hampered the experimental results, they must be taken into account when explaining how variants are generated and maintained in baculovirus populations.

iii). I think this point should be mentioned, as in previous papers there are references to “hot spots of mutations”.

The authors analyze the robustness of their model by a sensitivity analysis, but only of some of the parameters. It is interesting that the replication mechanism (strain copying or rolling circle) does not change the mutation rate estimates. What about the value of kappa, that is, the size of the virus population on a given host. Has a sensitivity analysis been performed using various estimates of the population size. Clearly the number of genomes present in the OBs is a subset of the total number of genomes replicated. It does not take into account the genomes that for BV not those that could be not included in nucleocapsids. It could be expected that the subset encapsidated into the is chosen randomly, but this could be not the case as OB production occurs at later times of infection, and thus the cellular machinery of repair might be not in optimal conditions. In a similar way, the multiplicity of infection, that is, the value of parameter Lambda, has been fixed to 46, due to the mode of infection of the larvae. The authors have fixed the numbers of ODV by OB, and fixed the number of nucleocapsids per ODV, but these values are very variable, even whithin a single production. The distribution of nucleocapsids par virion does not follow a normal law. A high number of ODV have a single nucleocapsid. Can the authors speculate on the consequences of varying this parameter, as the final number of genomes entering into the larvae is not known?

Minor remarks

Line 132. What does it means "per strand copying, the number of replication cycles in each insect is not known.

Line 154; What means “viral generation?

Line 193. As you provided evidence that nonsynonymous mutations are cleaned from the genomes, why not to use the synonymous mutations only in viral genes when calculating the mutation rate? This could help on the analysis of the transition to transversion ratio (line 236).

Line 303. The value of Lambda should be -10*ln(1-0.99) to get a positive value.

Reviewer #2: The authors present a concise examination of the mutation rate of a particular BAC-cloned isolate of Autographa californica multiple nucleopolyhedrovirus (AcMNPV), after experimental evolution for five passages in Spodoptera exigua (beet armyworm) larvae. The study is carefully executed from a bioinformatics and modeling perspective. The included code examples are well-annotated and documented. However there is a notable lack of consideration for how the biological context of the experiment may have influenced the outcome, as well as an over-concentration on comparisons to constraints that are particular to similar studies of RNA viruses.

Major issues

- One biological context that goes unmentioned in this study is the matchup of the particular AcMNPV BAC clone under study, and the Spodoptera exigua (beet armyworm) larvae used as the experimental evolution host. Presumably there was a rationale for choosing this isolate and this host – can the authors please add this? Might the choice of host be influencing the level of selective pressure on the virus? The discussion is extremely short, so there is room to explain how these choices may have influenced the outcome.

- Another biological context that goes undiscussed is the choice of an extremely high MOI inoculum – despite the low stated bottleneck of 46 – and the experimental design that pools billions of viral particles across multiple larvae cadavers, to create the next round of viral inoculum for each lineage. This approach seems sure to reduce the impact of any rare variants, and does not seem likely to reflect natural environmental bottlenecks. This leaves the impression that while the mutation rate can be as low as that measured here, the natural situation may be more varied.

- Two other aspects of the natural biological context for these viruses are standing genetic variation in the population, and the ability of coincident viruses to undergo recombination. The authors have intentionally aimed to use a homogeneous source population, but these other contributions impact the natural polymorphisms observed in isolates of AcMNPV at least belong in the discussion. At present, the authors leave it thus: "By contrast, the high genetic variation often observed within alphabaculovirus populations [21-24] remains a conundrum." The biological contexts listed here, among others, would help to explain this conundrum.

- Throughout the text, there is an over-emphasis on comparing the authors' work to data, methods and controls that have been necessary in prior studies of small RNA virus genomes. The vast majority of literature cited for comparison is likewise on (very excellent) RNA virus studies. Yet these are not the best comparisons for the present study, and this emphasis is likely to confuse the reader. In the introduction, the authors posit that "most mutations in viral genomes are deleterious", with citations to RNA virus studies. Large DNA viruses have sufficient intergenic space, genetic redundancy, repetitive elements, etc. that this claim is not likely to be correct for these viruses. This claim should be removed or backed up with relevant DNA virus references. As a methods example, the CirSeq approach used as a comparison on line 270 is an approach developed for RNA viruses to counterbalance for the need to use (error-prone) reverse transcriptase to create DNA templates from the initial pool of RNA virus genomes. That biological step is not necessary for baculoviruses, so this is not a useful contrast to make with the authors' methods here. It would be more useful and appropriate to compare the present work to recent in-depth genomics analyses of adenoviruses, poxviruses, herpesviruses, megaviruses, etc.

- The comparisons in the manuscript focus on the mutation rate of the AcMNPV genome, as compared to the "neutral intergenic region" of the BAC-insertion. While this is interesting, it seems odd to not include any mention or analysis of the natural intergenic regions of the AcMNPV genome. Since the authors calculated dN/dS, their analysis pipeline includes an awareness of gene-encoding regions and the intergenic areas between them. At present it is not stated how these are handled. Did the mutations observed in this study all fall into coding regions? Or were they predominantly intergenic? These data should be included, and the natural AcMNPV intergenic regions should be analyzed as a group for comparison, alongside the artificial neutral-intergenic (dI) region of the BAC, and the dN/dS rate of the rest of the AcMNPV genome.

- It is not clear why the authors chose to exclude the sizable number of mutations that occurs in the ancestral BAC, and any that showed up in more than one progeny virus. While the text (lines 313-316) suggests that this was done to exclude any contribution of variants in the initial starting population, this seems like an odd choice – since natural virus populations may have this much or an even larger amount of variation. The choice to exclude variants that appear more than once likewise seems poised to exclude the very real possibility of mutational hotspots in the viral genome.

Minor issues

- line 265 - this approach is not particularly novel. There is a long and classic literature using the insertion of exogenous sequences into viral genomes (and many other species), and using various methods to observe the mutation rate of the inserted sequence. Recognition and citation of this history would be more appropriate, and then the authors can distinguish how their application of this deep sequencing and this particular host-virus matchup provides novelty.

- What isolate or lineage of AcMNPV is contained in pBac-E2? Was this resource generated for this study, or is there a prior citation or description of its construction? These details should be referenced or included.

- How were the library preps generated for HiSeq sequencing? Presumably PE150 indicates paired-end reads?

- Why not include other large DNA viruses in figure 4? There are relevant comparative data on from similar studies of adenoviruses, poxviruses, herpesviruses, megaviruses, etc. These should be included as well.

- What is observed around the homologous regions (hrs) of this baculovirus genome? Does the alignment approach (mapping to the reference genome, line 308) obscure the ability to observe any fluctuations at these repetitive regions?

- What reference genome was used here, for mapping of all data (line 308)? Does it have an accession number in a common repository (EBI/GenBank)? It is not stated if it is derived from exactly this isolate, or if not, how closely it matches the BAC.

- Figures S1-S3 could easily be combined into a single figure.

- Sequencing data should be deposited in a common repository such as the Sequence Read Archive.

Reviewer #3: PGENETICS-D-21-01205

Boezen et al

This manuscript describes an in vivo experiment designed to calculate the mutation rate of the baculovirus AcMNPV based on mutation accumulation. The design if the experiment is rather elegant as the construction of a bacmid allowed both to start the experiment from a genomic clone and achieve normal in vivo replication in the caterpillar host Spodoptera exigua. The authors estimate a mutation rate of 10-7 both on the neutral inserted portion of the genome as well as the original portion of the genome. This is a rather conservative estimate and I wonder if the number of assumptions made truly reflects what could happen in natural populations.

Main comments:

L 109: Isn’t the viral amplification step in L3 a 6th passage in the experiment. Please explain the possible impact this amplification passage could have on the parameters used for the models to estimate the mutations rates

L113: Around there indicate the average sequence 5500 coverage for your genome. It is much lower than in previous deep sequencing analyses done on AcMNPV.

L258-259: In the conclusion it would help readers to estimate how many neutral mutations per genome (and per OB) could be transmitted to the next generation given the actual calculated rate. This could be compared this with the number of transposable elements found in AcMNPV OBs.

L286: What was the volume of the droplet used for the droplet feeding assay. (How many OBs do the caterpillar ingest?). How does this relate to the genome bottleneck size of 46?

L291: What was the OB concentrations and volumes used of infect the L3 caterpillar in the 6th (amplification) cycle? Was the final size of the virus population per insect cadaver still 5.32x108 ? In L3 the yield should be higher than in L2.

L312-317: I don’t understand why convergent evolution of mutation in different lineage should be excluded as possibly deriving from polymorphism in the BAC population. Is this an indication of insufficient sequence coverage for the experiment?

L 320: Would the sequencing error rate in the illumina data allow to use a lower mutation threshold given the sequencing coverage?

L330-332: what would it change if reversions are allowed?

L387-389: Throughout the material and methods the authors assume mutations are extremely rare and seem to exclude all data and parameters that would increase the calculated rate.

**Have all data underlying the figures and results presented in the manuscript been provided?**

Reviewer #1: Yes

Reviewer #2: **No: **Sequencing data should be deposited in a common repository such as the Sequence Read Archive.

Reviewer #3: **No: **the sequence data does not seem to be available as :' Sequence data have been uploaded to a permanent repository within the Netherlands Institute of Ecology and are available upon request.'

PLOS authors have the option to publish the peer review history of their article (what does this mean?). If published, this will include your full peer review and any attached files.

Reviewer #1: **Yes: **Miguel Lopez-Ferber

Reviewer #2: No

Reviewer #3: No

---

## [Decision Letter · Decision Letter 1]

27 Apr 2022

Dear Dr Zwart,

We are pleased to inform you that your manuscript entitled "Empirical estimates of the mutation rate for an alphabaculovirus" has been editorially accepted for publication in PLOS Genetics. Congratulations!

Yours sincerely,

Harmit S. Malik

Associate Editor

PLOS Genetics

Bret Payseur

Section Editor: Evolution

PLOS Genetics

Comments from the reviewers (if applicable):

Reviewer's Responses to Questions

**Comments to the Authors:**

Reviewer #1: In this revised version, I found the authors have considered the comments of the reviewers in a satisfactory way.

Line 225. Strictly speaking, there is not a single neutral insertion of 11.6 kbp, but two regions separated by a relatively highly selected gene, the polyhedrin, that conditions between host survival.

Reviewer #2: The authors have addressed the reviewers concerns well, and the revisions to the manuscript have made it more thorough and more clear.

Reviewer #3: I am satisfied with the response to reviewer comment provided. I particularly appreciate the substantial effort made to revise the molecular evolution model to address the different points made by the 3 reviewers and ensuing discussion.

I noted one typo, but there might be others:

l376: remove one 'we'

**Have all data underlying the figures and results presented in the manuscript been provided?**

Reviewer #1: Yes

Reviewer #2: Yes

Reviewer #3: Yes

PLOS authors have the option to publish the peer review history of their article (what does this mean?). If published, this will include your full peer review and any attached files.

Reviewer #1: **Yes: **Miguel LOPEZ-FERBER

Reviewer #2: No

Reviewer #3: No

**Data Deposition**

http://datadryad.org/submit?journalID=pgenetics&manu=PGENETICS-D-21-01205R1

**Press Queries**

---

## [Editor Report · Acceptance letter]

31 May 2022

PGENETICS-D-21-01205R1 

Empirical estimates of the mutation rate for an alphabaculovirus 

Dear Dr Zwart, 

We are pleased to inform you that your manuscript entitled "Empirical estimates of the mutation rate for an alphabaculovirus" has been formally accepted for publication in PLOS Genetics! Your manuscript is now with our production department and you will be notified of the publication date in due course.

With kind regards,

Zita Barta

PLOS Genetics

On behalf of:
